# The Effect of Municipal Solid Waste Incineration Ash on the Properties and Durability of Cement Concrete

**DOI:** 10.3390/ma15134486

**Published:** 2022-06-25

**Authors:** Marija Vaičienė, Elvinas Simanavičius

**Affiliations:** Civil Engineering Faculty, Vilnius College of Technologies and Design, 10303 Vilnius, Lithuania; elvinas.simanavicius@stud.vtdko.lt

**Keywords:** cement concrete, municipal solid waste incineration ash, workability, physical and mechanical properties, concrete durability, freeze–thaw resistance

## Abstract

The aim of this study is to investigate the effect of municipal solid waste incineration bottom ash from a cogeneration plant on the physical and mechanical properties and durability of cement concrete. Part of the cement in concrete mixtures tested was replaced with 0%, 3%, 6%, 9%, and 12% by weight of municipal solid waste incineration bottom ash. Concrete modified with 6% of bottom ash had a higher density (2323 kg/m^3^), compressive strength at 28 days (36.1 MPa), ultrasonic pulse velocity (3980 m/s), and lower water absorption rate (3.93%). The tests revealed that frost resistance, determined in all-sided testing directions, of concrete modified with 6%, 9%, and 12% of bottom ash added by weight of cement corresponds to strength grade F100. Such concrete can be used in construction works.

## 1. Introduction

The concept of a circular economy highlights the minimisation of disposed waste by recycling it in various industrial processes. The recycling of bottom ash from the incineration of municipal solid waste (MSW) in cement-based materials is one of the trends in the development of a circular economy. The use of MSW has a unique potential to support sustainability while preserving natural resources. Greenhouse gas (GHG) emissions in cement manufacturing are very high, accounting for about 8% of the global carbon dioxide (CO_2_) emitted [1,2]. The cement industry contributes to more than 85% of CO_2_ and GHG emissions [3]. The use of bottom ash can reduce the cement content in concrete and at the same time moderate environmental pollution and CO_2_ emissions.

As waste reuse and recycling are more expensive alternatives to landfilling or energy recovery, 17.6% of municipal waste is still landfilled in Lithuania [4]. However, the landfilling of waste causes social, economic, and environmental problems [5,6]. Waste incineration process generates bottom ash (slag) and steam boiler dust (boiler ash), which are classified as non-hazardous waste. Bottom ash has a high potential for reuse in cement-based products as a partial cement replacement, thus reducing the ash content in landfills and the cement content in cement products [7].

Fly ash generated in various industries [8,9,10] have been used in the production of concrete mixtures for many years [11,12,13]. In the last decade, the use of ash as a fine and/or coarse aggregate in concrete has received considerable attention in various countries in order to widen the range of potential uses of fly ash in the construction sector [14,15,16]. Researchers [17] tested the possibility of replacing 100% of natural sand in mortars with 0/2 mm MSW bottom ash. The leaching, physical, and chemical properties of the ash were analyzed. The results of the produced cement mortars showed that the compressive strength of the specimens containing ash ranged from 19.4 MPa (without superplasticizer) to 33.3 MPa (with 3% of superplasticizer). Huynh and Ngo [18] investigated the potential re-utilization of bottom ash as a replacement for crushed sand in cement-based mortars. It was found that the use of ash at higher levels (0%, 25%, 50%, 75%, and 100%) increases porosity, water absorption, and drying shrinkage, and reduces the compressive and flexural strength of the mortar specimens. After 56 days of curing, all mortars showed very good ultrasonic pulse velocity test results, with values ranging from 3483 to 4416 m/s. Authors [19] described the substitution of sand with 0%, 10%, 20%, and 30% of ash in cementitious mixes. The partial replacement of sand by ash increased the early (3-day) compressive strength of the cement mortar. The optimal results were achieved at 10% and 20% replacement ratios. 

Great attention has been paid to the use of bottom ash as a secondary raw material for the production of Portland cement clinker [20,21,22]. One of the main reasons for using MSW bottom ash in the cement clinker manufacturing process is that chemical composition of bottom ash is similar to that of the raw cement clinker materials, including lime, silica, alumina, iron oxide/hematite, and calcium sulphate [23]. Municipal solid waste incineration ash can partially replace Portland cement in concrete. Zhang et al. [24] tested the replacement of cement with 15% bottom ash in concrete mixes. The results show that concrete specimens have much better strength and durability values when activated by carbonation. Dry-cast concrete with 20% cement replaced by bottom ash showed 18% higher strength [25]. Cheng et al. [26] found that the optimal ash content in concrete was between 10% and 15%. Within this range, the ash not only improves the durability of concrete, but also ensures that concrete strength meets the design requirements. Other researchers [27] recommend replacing up to 15% of cement with ash to achieve low strength concrete. Although bottom ash has a porous structure, good pozzolanic properties of the ash were observed after grinding and thus bottom ash can be used as supplementary cementitious materials.

The composition of bottom ash depends on the incineration technology and the type of solid waste. The bottom ash consisted of silica, calcium, alumina, and iron [28]. Aluminium leads to the formation of aluminium hydroxide and high porosity of concrete due to the release of hydrogen in the reaction with water [29,30,31]. MSW incineration bottom ash, when added to the concrete mix directly, causes swelling and cracking due to the reaction between cement and metallic aluminium [32]. To avoid early age in concrete flooring, steel and polypropylene fibres are added to concrete mixes [33]. The main function of concrete fibres is to reduce the appearance and propagation of micro-cracks [34]. There is an increasing use of fibre-reinforced polymer composites as an efficient alternative to traditional construction materials in civil engineering and infrastructure applications [35,36]. Therefore, polypropylene fibres (PP), which do not undermine the durability of concrete, were used in concrete mixes tested in this research work.

Considering the long-term prospects for the recycling of MSW ash, there is a clear trend in the replacement of traditional aggregates with bottom ash and fly ash in concrete mixtures. The analysis of the literature about the use of bottom ash in concrete shows that future research needs to consider in great detail the workability, tensile strength, and drying shrinkage characteristics of concrete modified with bottom ash.

The aim of this study is to investigate the effect of bottom ash, generated from a cogeneration plant incinerating municipal solid waste, on the physical and mechanical properties and durability of cement concrete in terms of frost resistance, and to assess the possibility of replacing part of the cement in the mixes with the MSW incineration ash. This concrete can be used for industrial flooring applications.

## 2. Materials and Methods

### 2.1. Raw Materials

The following binding materials were used: commercial Portland cement CEM I 42.5 R (containing 95–100% of clinker) produced by AB Akmenės cementas, Naujoji Akmenė (Lithuania); and bottom ash from a cogeneration power plant UAB Gren Lietuva, Klaipėda (Lithuania). The chemical composition of the cement and bottom ash (BA) is given in Table 1 and the physical and mechanical properties of cement are given in Table 2. The mineral composition of the cement was as follows: 62.5% of C_3_ S, 16.9% of C_2_ S, 7.1% of C_3_ A, 11.5% of C_4_ AF, and 2% of other materials (alkali sulphates and CaO). BA was obtained from the cogeneration plant, where municipal waste is incinerated at a temperature of ~1100 °C. The ash was stored for one year in the Klaipėda regional landfill in Dumpiai under natural conditions. Afterwards, it was dried at 105 °C and ground in a ball mill. BA had the bulk density of 0.91 g/cm^3^ and the particle fineness (residual on a 90 μm sieve) of 4.0%. The bottom ash is attributed to Class F, as the total SiO_2_ + Al_2_O_3_ + Fe_2_O_3_ content exceeds 70%.

0/2 fraction sand with the particle size distribution given in Figure 1 (EN 933-1) was used as a fine aggregate in the tests. The physical properties of the sand are given in Table 3.

5/16 fraction granite crushed stone (bulk density 1360 kg/m^3^) was used as a coarse aggregate in the tests (50%—5/8 fraction and 50%—11/16 fraction).

Polypropylene fibre (PP) Crackstop F6 Adfil NV (Belgium) with a density of 0.905 kg/dm^3^, equivalent diameter of 0.78 mm, length of 6 mm, and tensile strength of 110 MPa measured according to EN 14889-2 was used to prepare the concrete.

Superplasticiser Master Glenium ACE 560 (SP) MBCC Group, Master Builders Solutions (Poland), was used in the mixes. It is a high-performance liquid admixture based on the new generation of polycarboxylate ether polymers, with a density of 1.06 g/cm^3^ and pH value 5.6 at T = 20 °C.

Tap water was used to prepare the samples.

### 2.2. Mix Design and Sample Preparation

The cement content in the concrete mixes varied up to 12%. The maximum ash content was chosen by taking into account the data already presented by other authors [37], who pointed out that 5% and 10% of ash was the optimal amount to be added to cement mortars. The compositions of concrete mixes with bottom ash are given in Table 4. It can be seen that the ash to binder ratios of concrete mixes differ. The water to binder ratio (V/B) 0.55 was kept constant for all the concrete mixes with ash tested. The superplasticizer was added at 1% of the cement content.

15 specimens were cast for each concrete mix. Concrete cubes with the dimensions of 100 × 100 × 100 mm were cast in reusable stainless-steel moulds. Concrete pastes were compacted on the laboratory vibrating table (MATEST S.p.A., Arcore, Italy). The formed specimens were stored for one day in covered moulds, then demoulded and cured according to LST EN 12390-2. Three specimens from each batch were used to test the durability, and the physical and mechanical properties of concrete.

### 2.3. Testing Methods

The compressive strength of concrete specimens was determined according to EN 12390-3 with a hydraulic press ALPHA 3-3000S test machine (FORM+TEST Seidner + Co. GmbH, Riedlingen, Germany). Three specimens of each mix were tested at 7 and 28 days of curing. The density of hardened concrete was determined according to EN 12390-7, the density of compacted fresh concrete was determined according to EN 12350-6, the slump was determined according to EN 12350-2, and the flow was determined according to EN 12350-5. Ultrasonic pulse velocity was determined using a Pundit 7 instrument (converter frequency = 54 kHz) calculating the ultrasonic pulse velocity (UPV, m/s) according to the literature [38]. To determine the water absorption kinetics [39], the specimens were dried to a constant mass, weighed, immersed in water, and weighed in air after 10 min, 30 min, 60 min, 24 h and 48 h.

The porosity parameters of concrete were determined by measuring water absorption kinetics according to the methodology presented in GOST 12730.4-78. This method is used to describe the open porosity (*P_a_*) (capillary pores), the total porosity (*P_t_*), and the closed porosity (*P_u_*) (air pores) of concrete. The open porosity was calculated from the following Equation (1):(1)$$P_{a}=\frac{W\cdot \rho }{\rho_{w}},\;\%$$
where *W* is water absorption after 48 h, %; *ρ* is the density of concrete, g/cm^3^, *ρ_w_* is the density of water, g/cm^3^. The total porosity was calculated from the following Equation (2):(2)$$P_{t}=\left(1-\frac{\rho_{t}}{\rho_{p}}\right)\cdot 100\%$$
where *ρ_t_* is the density of concrete, kg/m^3^; *ρ_p_* is the specific density of concrete, kg/m^3^.

The closed porosity of concrete was calculated from the following Equation (3):*P_u_ =P_t_ − P_a_, %*(3)

The frost resistance of concrete using the all-sided testing direction was tested using the Rumed 3301 climate chamber (Rubarth Apparate GmbH, Laatzen, Germany) in order to determine the effect of freezing and thawing (FT) cycles on the compressive strength of concrete when different amounts of fly ash were used in concrete mixes. The durability of concrete specimens was determined after 28 days of curing according to the requirements of LST 1428-17. According to the accelerated frost resistance test, water saturated specimens are subjected to cyclic freezing in air and thawing in water. For the accelerated test, the specimens are soaked in 3% aqueous sodium chloride solution (NaCl). The frost resistance class F100 corresponds to the resistance to 20 FT cycles. Prior to the FT test, the saturated specimens are removed from the water bath and placed in such a manner that the water can drain from them. After 2–4 h following the removal of the specimens from the water bath, the control specimens are tested for their initial compressive strength according to LST EN 12390-3. The remaining concrete specimens are placed in the freezing chamber. The specimens are air frozen for at least 2.5 h. The temperature in the centre of the freezing chamber are (−18 ± 2) °C during freezing. After determining the frost resistance, the specimens are removed from the freezing chamber and placed in the bath of aqueous NaCl salt solution at (18 ± 5) °C. The specimens are kept in the bath for at least (2 ± 0.5) h. The all-sided frost resistance of concrete specimens is determined not only by measuring the compressive strength but also by the mass loss. If, after the required number of FT cycles, the loss in compressive strength of concrete specimens does not exceed 5% and/or mass loss of the concrete specimens does not exceed 3%, the concrete is considered to have passed the frost resistance test. Two freeze–thaw cycles per day were carried out with the tested specimens. After 100 cycles, the specimens were crushed and the loss in strength and mass was calculated.

## 3. Results and Discussion

### 3.1. Fresh Concrete Properties

Slump and flow tests were carried out in order to determine the effect of bottom ash on the consistency of the fresh concrete. The results obtained using a constant water/binder ratio of 0.55 are shown in Table 5. It is observed that the workability of the fresh concrete decreases with a higher ash content. The flow and slump values decreased because BA has a higher water absorption rate, a bigger natural air content, and a lot of small particles that have a negative effect on the consistency of fresh concrete [40]. A similar effect of ash on the consistency of concrete was reported by other authors [41,42] who stated that the average water absorption of ash can be as high as 9.3%, thus deteriorating the workability of fresh concrete. The average density of the fresh concrete was found to be in the range of 2343–2425 kg/m^3^. As expected, the density of fresh concrete decreased due to the loss of internal moisture [43]. All cement pastes of the tested concrete mixes corresponded to slump class S2 with the slump range between 50–90 mm. Concrete mixes B0, B3, and B6 had the flow diameter of more than 410 mm; therefore, these mixes can be classified under consistency class F3.

### 3.2. Properties of Hardened Samples

#### 3.2.1. Density and UPV

The properties of hardened concrete depend on the constituents of the concrete mix and their proportions. When bottom ash is used in the concrete mix, it is important to determine the effect of ash on the properties of hardened concrete. In this work, concrete specimens were tested by changing the ash content in the mix. Control concrete specimens without ash were used as the reference. The addition of different amounts of fly ash to the mix resulted in concrete that varied not only by density but also by compressive strength values.

The results given in Figure 2 show that the density of concrete specimens containing different amounts of ash vary from 2264 to 2323 kg/m^3^. The average density values obtained in the experimental tests vary within a narrower range: the average density of specimens B0 was 2318 kg/m^3^, the average density of specimens B3 was 2264 kg/m^3^, the average density of specimens B6 was 2323 kg/m^3^, the average density of specimens B9 was 2271 kg/m^3^, and the average density of specimens B12 was 2315 kg/m^3^. It should be noted that specimens B6 had a denser structure because their density values were the highest. The obtained results show that the addition of bottom ash at 3% causes a 2.3% drop in density. When 9% of ash is added into the mix, the drop in density becomes less significant at only 2.0%. The tests results show that bottom ash does not have a significant effect on the density of concrete. Similar results were reported by Abubakar [44]; the replacement of cement with 5% and 10% with bottom produced the density values 2313 and 2323 kg/m^3^, respectively.

The variation in the ultrasonic pulse velocity (UPV) values in relation to the ash content is shown in Figure 3. After 28 days, the ultrasonic pulse velocity varied between 3856 and 3980 m/s. With the increase in ash content up to 6%, a 3% increase in the ultrasonic pulse velocity was observed. It should be noted that in the specimens containing 9% and 12% ash, the ultrasonic pulse velocity was slightly reduced compared to specimens B6. According to Lafhai et al., the porosity of the cement mortar is the main factor affecting the UPV value, i.e., higher porosity has a negative effect on UPV [45]. The mortars tested by researchers [18] obtained similar UPV values ranging from 3483 m/s.

The UPV is the most consensual in situ and non-destructive test to evaluate the quality of concretes. Table 6 presents a classification available in the literature to determine the quality of concrete based on UPV values. The higher the ultrasonic pulse velocity (Table 6), the better the quality of the concrete. The variation of ultrasonic pulse velocity in concrete is also determined by its composition, density, water/cement ratio, curing conditions, fibre types, and aggregates [46,47,48].

#### 3.2.2. Compressive Strength Tests

In this work, the compressive strength of concrete specimens was determined after 7 and 28 days of curing. The average compressive strength values obtained in the experimental tests are shown in Figure 4. The results presented in Figure 4 show that the early strength of the specimens increases with a higher ash content. The control specimens have a compressive strength of 25.0 MPa after 7 days. The highest early compressive strength of 31.4 MPa was obtained for the specimens containing 6% of ash and cured for 7 days. The compressive strength of the specimens containing 9% and 12% of ash remain the same at 27.2 MPa. A negative effect of higher ash content on the early strength of concrete may be caused by delayed cement hydration because the CaO content is insufficient for the development of cement compounds [37].

The results of further tests show (Figure 4, after 28 days of curing) that the average compressive strength of the control specimens is 32.6 MPa. When cement was replaced with bottom ash in concrete mix, the average compressive strength results varied in quite a wide range: 32.6 MPa for specimens B0, 33.9 MPa for specimens B3, 36.1 MPa for specimens B6, 32.7 MPa for specimens B9, and 33.1 MPa for specimens B12. The highest average compressive strength values were obtained for the specimens where 6% of cement was replaced with bottom ash. Figure 4 shows that the compressive strength in the specimens containing 6% of bottom ash increases by 10.7%, compared to the control specimen. The increase in the strength of the specimens may be related to the denser structure produced by the products of cement hydration. The standard deviation in all batches is between 0.2% and 2.3%. The tests show that after 28 days of curing, the compressive strength of the specimens containing 9% and 12% of bottom ash decrease ~9%, compared to specimens B6. When part of the cement was replaced with bottom ash, the relative calcium aluminate content decreased. The finely ground bottom ash reduces the mechanical strength of the mortars due to very high porosity and susceptibility to cracking associated with metallic aluminium and the reduced reaction of alkaline silica [50].

A slight decrease in the compressive strength was also reported by Rafieizonooz [51]. Researchers [52] investigated the effect of partial replacement of Portland cement with bottom ash on the compressive strength of concrete. They replaced 30% of cement with dry ground bottom ash. After 28 days of curing, the compressive strength of these concrete specimens was 20.1 MPa. Other researchers [53] studied cement mortars in which 30% of cement was replaced with ground bottom ash. The results showed that after 28 days of curing, the compressive strength of the specimens was almost 40.0 MPa. The authors [26] replaced cement with 10% and 15% of bottom ash in concrete mixes. The studies show that after 28 days of curing, the compressive strength of the cement pastes with W/C ratio of 0.45 reached 29 MPa and 26 MPa, respectively. After 28 days of curing, the compressive strength of the cement pastes where cement was replaced with 10% of bottom ash was ~22 MP, showing a 4% decrease compared to the control specimens, which is not significant [43].

#### 3.2.3. Water Absorption Kinetics Test

To determine the water absorption kinetics in concrete, the specimens were dried to a constant mass, weighed, soaked in water, and weighed in air after 10 min, 30 min, 60 min, 24 h, and 48 h. The results of water absorption in the specimens cured for 28 days are given in Figure 5. The results of water absorption kinetics test show the progress of hardened cement paste hydration (the change in hydration). There is a great variation in the number of free particles that do not react during the hardening of the cement paste. It should be noted that water absorption in the specimens reached ~1% in the first 10 min, 1.6% after 30 min, and 2% after 60 min. A more detailed analysis of water absorption curves shows that the fastest water absorption takes place during the first 60 min. After 24 h of testing the specimens have absorbed 4% by weight. The water absorption trends remained the same after 48 h. The water absorption kinetics test show that the specimens containing 6% of ash had lower absorption values compared to other specimens. Hardened cement paste had a lower water absorption value due to its structure. This is due to the change in the structure of specimens B6 as a result of increased density. According to Powers, 5 μm sized cement particles hydrate over 1–3 days, whereas the particles with the size of 5–10 μm finish hydrating in 28 days. These differences in particle hydration time change the structure of hardened cement paste [54].

#### 3.2.4. Freeze–Thaw Resistance Test

The variation of the open and closed porosities of concrete containing ash is given in Table 7. This table shows that the open porosity of concrete changes only slightly when the ash content increases up to 12% by the weight of cement. The open porosity of the concrete varied between 9.1% and 9.6%, the total porosity varied between 14.0% and 16.2%, and the closed porosity varied between 4.7% and 6.9%.

Open pores and capillaries are formed when free water escapes from concrete. The number and size of such pores depends on the water/cement ratio. Closed pores are formed due to the entrainment of air from the environment and due to the contraction of the hardening cement paste [55].

The results of frost resistance test are given in Table 8. This table shows that not all concrete specimens passed the 100-cycle FT test. After 100 freeze–thaw cycles, the greatest drop in compressive strength in MPa is observed in concrete compositions B3, while in concrete compositions B6 a lower drop is observed. The analysis of compressive strength results reveals that the compressive strength of compositions B9 and B12 increased. Concrete specimens with the lowest fly ash content, i.e., the control specimens and the specimens containing 3% of ash by weight of cement, show the highest decrease in compressive strength after 100 cycles: 6.1% and 6.9%, respectively. In contrast, the specimens containing 6% of ash lost only 3.9% of their strength. No cracks were observed in concrete specimens after the durability test.

The decrease in compressive strength of the unbroken B6, B9 and B12 specimens after 100 freeze–thaw cycles was found to be within 5%. The mass loss for all concrete specimens did not exceed 3%. Based on the results obtained, specimens B6, B9, and B12 comply with the F100 frost resistance class.

## 4. Conclusions

The tests conducted in this research work reveal positive prospects for the use of bottom ash in concrete. According to the test results, MSW incineration bottom ash has an effect on physical-mechanical properties and durability in terms of frost resistance of concrete.

6% is the optimal content of bottom ash in concrete according to density and compressive strength values at 28 days. The density of the specimens modified with 6% of bottom ash is 2323 kg/m^3^, and the average compressive strength is 36.1 MPa. A higher content of bottom ash reduces the slump flow and density of the mix. The compressive strength of concrete specimens after 28 days of curing corresponds to the design strength class C25/30. The quality of the concrete is good, as the average ultrasonic pulse velocity in the specimens is 3912 m/s.

It was found that water absorption of concrete increased significantly up to 4.1% after 24 h. Concrete specimens modified with 6% of bottom ash had a lower water absorption rate of 3.9% after 24 h and 48 h.

In summary it can be stated that concrete modified with 6%, 9%, and 12% of bottom ash has good durability in terms of frost resistance and can be used in construction work where resistance of concrete to 100 freeze–thaw cycles (F100) is required.

## Figures and Tables

**Figure 1 materials-15-04486-f001:**
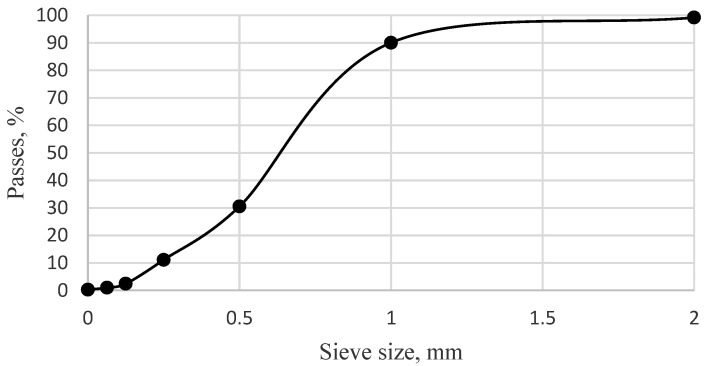
The particle size distribution of sand.

**Figure 2 materials-15-04486-f002:**
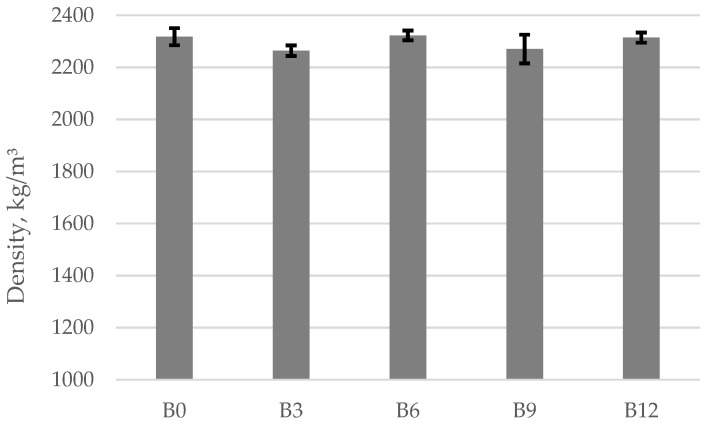
Average density values for concrete specimens containing ash.

**Figure 3 materials-15-04486-f003:**
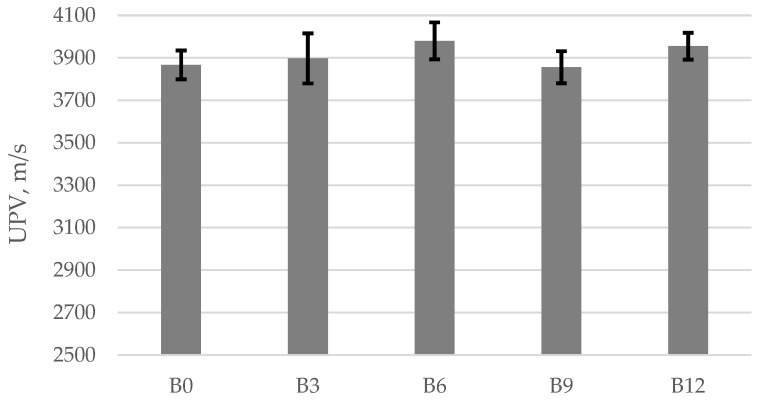
Ultrasonic pulse velocity.

**Figure 4 materials-15-04486-f004:**
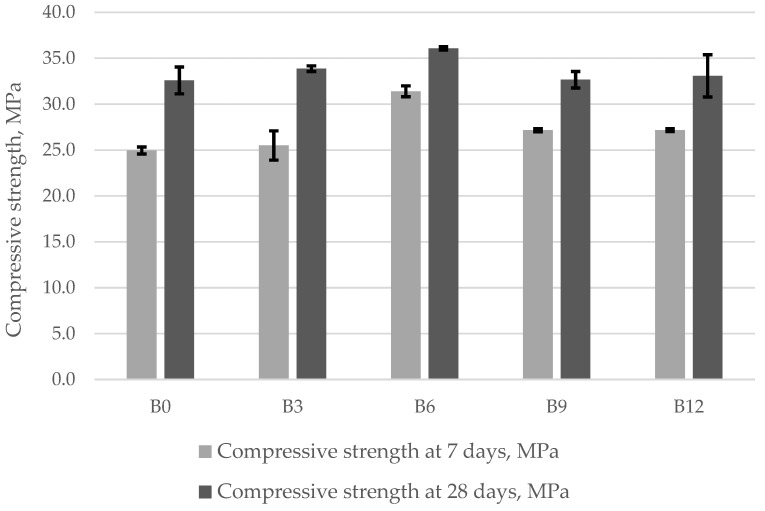
Average compressive strength values for concrete specimens containing ash.

**Figure 5 materials-15-04486-f005:**
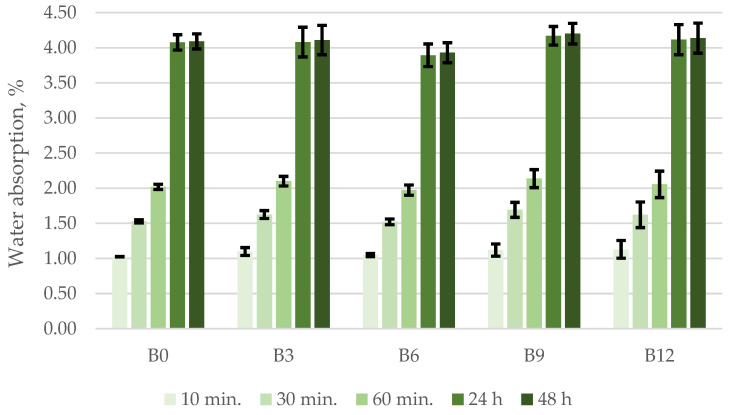
Water absorption kinetics of the concrete specimens.

**Table 1 materials-15-04486-t001:** Chemical composition of CEM I 42.5 R and BA (% by mass).

	CaO	SiO_2_	Al_2_ O_3_	Fe_2_ O_3_	MgO	K_2_ O	Na_2_O	SO_3_	Cl	Other
CEM I 42.5 R	63.2	20.4	4.0	3.6	2.4	0.9	0.2	3.1	0.05	2.2
Bottom ash (BA)	14.5	55.6	8.3	7.4	2.3	1.6	5.8	1.1	0.3	2.3

**Table 2 materials-15-04486-t002:** Physical and mechanical properties of CEM I 42.5 R.

	Particle Density, g/cm^3^	Bulk Density, g/cm^3^	Fineness, cm^2^/g	Compressive Strength after 2 Days, MPa	Compressive Strength after 28 Days, MPa	Initial Setting Time, min	Final Setting Time, min
CEM I 42.5 R	3.1	1.1	3700	20	55	160	215

**Table 3 materials-15-04486-t003:** Properties of the fine aggregate.

Fraction	Characteristics		
	Particle Density, kg/m^3^	Bulk Density, kg/m^3^	Amount of Fine Particles < 0.063 mm, %
0/2	2419	1635	0.98

**Table 4 materials-15-04486-t004:** Compositions of concrete mixes (kg/m^3^).

Mix Designation	Binders (Cement + BA)		Sand	Granite Crushed Stone	Water	BA, %	PP	SP
	Cement	BA						
B0	300	0	980	1000	165	0	0.9	3.0
B3	291	9	980	1000	165	3	0.9	3.0
B6	282	18	980	1000	165	6	0.9	3.0
B9	273	27	980	1000	165	9	0.9	3.0
B12	264	36	980	1000	165	12	0.9	3.0

**Table 5 materials-15-04486-t005:** Slump, flow, and density values of fresh concrete mixtures containing ash.

Mix Designation	Slump, mm	Flow, mm	Density, kg/m^3^
B0	68	449	2425
B3	67	440	2357
B6	65	418	2417
B9	59	400	2361
B12	55	395	2343

**Table 6 materials-15-04486-t006:** Concrete quality classification based on UPV values [49].

Concrete Quality	UPV (m/s)
Excellent	4500
Good	3600–4500
Questionable	3000–3600
Poor	2100–3000
Very poor	<2100

**Table 7 materials-15-04486-t007:** Concrete porosity parameters.

Mix Designation	Open Porosity, Pa, %	Total Porosity, Pt, %	Closed Porosity, Pu, %
B0	9.5	14.2	4.7
B3	9.3	16.2	6.9
B6	9.1	14.0	4.9
B9	9.5	16.0	6.5
B12	9.6	14.3	4.7

**Table 8 materials-15-04486-t008:** Freeze–thaw resistance results.

Mix Designation	B0	B3	B6	B9	B12
Mass loss, %	−0.17	−0.04	+0.42	+0.17	+1.30
Change in compressive strength, %	−6.1	−6.9	−3.9	+0.4	+3.6
Appearance of samples	No cracks observed				
Number of cycles	100				

## Data Availability

Not applicable.

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
