# Peer review of "The Effect of Municipal Solid Waste Incineration Ash on the Properties and Durability of Cement Concrete"

_materials, 2022, doi:10.3390/ma15134486_

Round 1

Reviewer 1 Report

Journal-Manuscript ID: materials-1752162-peer-review-v1

Title: The Effect of Municipal Solid Waste Incineration Ash on the Properties and Durability of Cement Concrete

 The main concerns are:

 - The experiment and the analysis of the results are extremely simple:

The work does not present a great novelty since it is in line with a great number of works already published in the subject. The experimental campaign is simple and the results are in line with what should be expected considering the used materials. The analysis of the results is also extremely simple. Moreover, what is the specific reason for using Polypropylene fibre (PP), which not mentioned any more in the section of "Results and Discussion"??

Author Response

Dear Reviewer, Thank You very much for comments. 

Response to Reviewer 1 Comments

Point 1: The work does not present a great novelty since it is in line with a great number of works already published in the subject. The experimental campaign is simple and the results are in line with what should be expected considering the used materials. The analysis of the results is also extremely simple. Moreover, what is the specific reason for using Polypropylene fibre (PP), which not mentioned any more in the section of "Results and Discussion"??

Response 1: Considering the long-term prospects for the recycling of MSW ash, there is a clear trend towards the replacement of traditional aggregates with bottom ash and fly ash in concrete mixtures. Fly ash generated in various industries have been used in the production of concrete mixtures for many years.  The analysis of literature has shown that there is less research in which part of the cement in concrete mixture was replaced with bottom ash. The results of this research are good, because by reducing the amount of cement in the concrete mixes, the properties and durability of the concrete do not deteriorate. The use of bottom ash can reduce the cement content in concrete and at the same time moderate the environmental pollution and CO2 emissions. MSW incineration bottom ash, when added to the concrete mix directly, causes swelling and cracking due to the reaction between cement and metallic aluminium. To avoid early age in concrete flooring, steel and polypropylene fibres are added to concrete mixes. The main function of concrete fibres is to reduce the appearance and propagation of micro-cracks. There is an increasing use of fibre-reinforced polymer composites as an efficient alternative to traditional construction materials in civil engineering and infrastructure applications. Therefore, polypropylene fibres (PP), which do not undermine the durability of concrete, were used in concrete mixes tested in this research work. So, this concrete can be used for industrial flooring applications. The aim of this study is to investigate the effect of the municipal solid waste incineration bottom ash from a cogeneration plant on the physical and mechanical properties and durability of cement concrete. Therefore, the section on "Results and Discussion" focuses on the effect of ash on concrete properties and durability. 

The introduction of the article was revised, and the list of references was increased by 22 sources. Materials and Methods was elaborated for every individual test (Raw Materials; Mix Design and Sample Preparation; Testing Methods). Result and Discussions was re-written in sub-headings each focusing on parameter measured (Fresh Concrete Properties; Properties of Hardened Samples; Density and UPV; Compressive Strength Tests; Water Absorption Kinetics Test; Freeze-Thaw Resistance Test).

Conclusions were corrected.

We have double-checked the grammar and improved the language.

The adjusted places in the article are marked in yellow.

Reviewer 2 Report

Introduction can be reduced and part of its content can be re-written as current research trend and significance of current research . 

Major headings can be sub divided into sub-headings for clarity and focus on specific content in those paras. Materials and Methods can be elaborated for every individual test. Results and Discussions can be re-written in sub-headings each focusing on specific parameter measured. Scientific reasoning for each of findings may be added to make paper more effective and interesting. Conclusions have to be made specific and point-wise with respect to each of measured parameters . 

Methology of the research can be explained in the form of an integrated flow chart etc. 

Researchers can add  micro structural testing findings , if done for making research more interesting and complete . 

Author Response

The authors would like to thank the reviewer for the time and valuable comments. Below, reactions to reviewer’s comments can be found.

Response to Reviewer 2 Comments

Point 1: Introduction can be reduced and part of its content can be re-written as current research trend and significance of current research 

Response 1: The introduction of the article was revised (an additional 16 reference sources were used for this purpose). 

Point 2: Major headings can be sub divided into sub-headings for clarity and focus on specific content in those paras. Materials and Methods can be elaborated for every individual test. Results and Discussions can be re-written in sub-headings each focusing on specific parameter measured. Scientific reasoning for each of findings may be added to make paper more effective and interesting. Conclusions have to be made specific and point-wise with respect to each of measured parameters

Response 2: Materials and Methods was elaborated for every individual test (Raw Materials; Mix Design and Sample Preparation; Testing Methods). Result and Discussions was re-written in sub-headings each focusing on parameter measured (Fresh Concrete Properties; Properties of Hardened Samples; Density and UPV; Compressive Strength Tests; Water Absorption Kinetics Test; Freeze-Thaw Resistance Test). Conclusions were corrected. Section "Results and Discussion" has been added with 6 reference sources.

Point 3: Methology of the research can be explained in the form of an integrated flow chart etc. 

Response 3: The second section "Materials and Methods" has been expanded and supplemented.

Point 4: Researchers can add  micro structural testing findings, if done for making research more interesting and complete

Response 4: The aim of this study is to investigate the effect of bottom ash, generated from a cogeneration plant incinerating municipal solid waste, on the physical and mechanical properties and durability of cement concrete in terms of frost resistance, and to assess the possibility of replacing part of the cement in the mixes with the MSW incineration ash. Therefore, microstructural testing were not performed

Conclusions were corrected.

We have double-checked the grammar and improved the language.

The adjusted places in the article are marked in yellow.

Reviewer 3 Report

The article "The Effect of Municipal Solid Waste Incineration Ash on the Properties and Durability of Cement Concrete" presents a comprehensive standard study of concrete mortars with the use of municipal solid waste incineration ash. An attempt has been made to conduct the research in such a way that it could be directly applied in practice. Satisfactory results were obtained with respect to ash content in concrete mortar up to 12 %, i.e. class C25/C30 was obtained.

Author Response

Response to Reviewer 3 Comments

Point 1: The article "The Effect of Municipal Solid Waste Incineration Ash on the Properties and Durability of Cement Concrete" presents a comprehensive standard study of concrete mortars with the use of municipal solid waste incineration ash. An attempt has been made to conduct the research in such a way that it could be directly applied in practice. Satisfactory results were obtained with respect to ash content in concrete mortar up to 12 %, i.e. class C25/C30 was obtained.

Response 1: The authors would like to thank the reviewer for the time and positive feedback.

We have double-checked the grammar and improved the language.

Reviewer 4 Report

This work is interesting where bottom ash has the possibility of replacing some part of the cement in concrete mixes. Besides, it has a good potential helping in waste management. The authors studied the effects of bottom ash in replacing some part of the cement in concrete. It is also interesting to see different bottom ash loading produced different results and performance of the concrete. However, there are some issues that should be addressed before publication

1. Please state the class of the bottom ash that you selected in your research.

2. In page 2, line 74 and line 77, it is suggested to put the authors name instead of Canadian and Chinese researchers.

3. In Page 6, line 223, it is suggested to put standard deviation for the graph.

4. In Page 7, line 260, please explain why addition of 9% and 12% ash for curing 7 and 28 days showed the same compression strength.

5. It is suggested to add micrographs of the concrete (microstructure analysis) to support the existing results.

Author Response

The authors would like to thank the reviewer for the time and valuable comments and positive feedback. Below, reactions to reviewer’s comments can be found.

Response to Reviewer 4 Comments

Point 1: Please state the class of the bottom ash that you selected in your research.

Response 1: In the subsection 2.1. Raw Materials was added text: The bottom ash is attributed to Class F, as the total SiO2 + Al2O3 + Fe2O3 content exceeds 70%.

Point 2: In page 2, line 74 and line 77, it is suggested to put the authors name instead of Canadian and Chinese researchers.

Response 2: In the Introduction was added text: Huynh and Ngo [18] investigated the potential re-utilization of bottom ash as a replacement for crushed sand in cement-based mortars. Zhang et al. [24] tested the replacement of cement with 15% bottom ash in concrete mixes. The results showed that concrete specimens had much better strength and durability values when activated by carbonation. Dry-cast concrete with 20% cement replaced by bottom ash showed 18% higher strength [25]. Cheng et al. [26] found that the optimal ash content in concrete was between 10% and 15%. 

Point 3: In Page 6, line 223, it is suggested to put standard deviation for the graph.

Response 3: In all graphs (Figure 2 - Figure 5) was added statistical analysis.

Point 4: In Page 7, line 260, please explain why addition of 9% and 12% ash for curing 7 and 28 days showed the same compression strength

Response 4: In the subsection 3.2.2. Compressive Strength Tests was added text: A negative effect of higher ash content on the early strength of concrete may be caused by delayed cement hydration because CaO content is insufficient for the development of cement compounds [37]. 

The increase in the strength of the specimens may be related to the denser structure produced by the products of cement hydration. The standard deviation in all batches was between 0.2% and 2.3%. The tests showed that after 28 days of curing, the compressive strength of the specimens containing 9% and 12% of bottom ash decreased ~9% compared to specimens B6. When part of the cement was replaced with bottom ash, the relative calcium aluminate content decreased. The finely ground bottom ash reduces the mechanical strength of the mortars due to very high porosity and susceptibility to cracking associated with metallic aluminium and the reduced reaction of alkaline silica [50].

Point 5: It is suggested to add micrographs of the concrete (microstructure analysis) to support the existing results.

Response 5: The aim of this study is to investigate the effect of bottom ash, generated from a cogeneration plant incinerating municipal solid waste, on the physical and mechanical properties and durability of cement concrete in terms of frost resistance, and to assess the possibility of replacing part of the cement in the mixes with the MSW incineration ash. Therefore, microstructural test were not performed.

The adjusted places in the article are marked in yellow.

Reviewer 5 Report

The manuscript presents the results of an experimental study about the effect of bottom ash from a cogeneration plant fired by the residual waste remaining after sorting on the physical, mechanical and durability properties of concrete. The text is very well written and few suggestions are made to improve the paper:

- The abstract could include a brief contextualization on the topic.

- Authors should include the particle size distribution of the coarse aggregate in the same figure as the fine aggregate.

- How many samples were used for each test? Authors should include this information in the methods.

- All hardened concrete test results are presented without statistical analysis. Please include error bars and standard deviation of the results.

- The conclusions are very summarized. Please rewrite giving more information about the results obtained.

Author Response

The authors would like to thank the reviewer for the time and valuable comments and positive feedback. Below, reactions to reviewer’s comments can be found.

Response to Reviewer 5 Comments

Point 1: The abstract could include a brief contextualization on the topic

Response 1: It was corrected. In the Abstract we wrote: The aim of this study is to investigate the effect of the municipal solid waste incineration bottom ash from a cogeneration plant on the physical and mechanical properties and durability of cement concrete. Part of the cement in concrete mixtures tested was replaced with 0%, 3%, 6%, 9%, and 12% by weight of municipal solid waste incineration bottom ash. Concrete modified with 6% of bottom ash had higher density (2323 kg/m3), compressive strength at 28 days (36.1 MPa), ultrasonic pulse velocity (3980 m/s), and lower water absorption rate (3.93%). The tests revealed that frost resistance, determined in all-sided testing direction, of concrete modified with 6%, 9%, and 12% of bottom ash added by weight of cement corresponds to strength grade F100. Such concrete can be used in construction works.

Point 2: Authors should include the particle size distribution of the coarse aggregate in the same figure as the fine aggregate.

Response 2: Since the coarse aggregate was purchased from the manufacturer fractionated in the article we wrote: 5/16 fraction granite crushed stone (bulk density 1360 kg/m3) was used as a coarse aggregate in the tests (50% - 5/8 fraction and 50% - 11/16 fraction).

Point 3: How many samples were used for each test? Authors should include this information in the methods.

Response 3: In the article was added text: 15 specimens were cast for each concrete mix. Three specimens from each batch were used to test the durability, physical and mechanical properties of concrete.

Point 4: All hardened concrete test results are presented without statistical analysis. Please include error bars and standard deviation of the results.

Response 4: In Figure 2 - Figure 5 we include statistical analysis.

Point 5: The conclusions are very summarized. Please rewrite giving more information about the results obtained.

Response 5: Conclusions were corrected.

We have double-checked the grammar and improved the language.

The adjusted places in the article are marked in yellow.

Author Response

The authors would like to thank the reviewer for the time and valuable comments. We think that all remarks were useful and quality of article was improved. Please see the attachment as well.

Round 2

Reviewer 4 Report

This work is interesting where bottom ash has the possibility of replacing some part of the cement in concrete mixes. Besides, it has a good potential helping in waste management. The authors studied the effects of bottom ash in replacing some part of the cement in concrete. It is also interesting to see different bottom ash loading produced different results and performance of the concrete. However, it would be better if the authors can add one more experiment (microstructure analysis) to support your current results.

Author Response

The authors would like to thank the Reviewer for the time and valuable comments and positive feedback. 

Response to Reviewer 4 Comments

Point 1: This work is interesting where bottom ash has the possibility of replacing some part of the cement in concrete mixes. Besides, it has a good potential helping in waste management. The authors studied the effects of bottom ash in replacing some part of the cement in concrete. It is also interesting to see different bottom ash loading produced different results and performance of the concrete. However, it would be better if the authors can add one more experiment (microstructure analysis) to support your current results.

Response 1: The main objective of this research was to determine the effects of bottom ash on the properties and durability of cement concrete. We studied the fresh concrete properties, properties of hardened samples (density, ultrasonic pulse velocity, compressive strength, water absorption kinetics and freeze-thaw resistance). Therefore, no microstructure analysis were performed and we do not have the possibility to perform this experiment.

Author Response

The authors would like to thank the Reviewer for the time and valuable comments and positive feedback. 

Response to Reviewer 6 Comments

Point 1: All major comments were adequately addressed and the Authors have done an admirable job of improving the quality of the manuscript. Notwithstanding, the Authors may still wish to address these final (very) minor points. I would still recommend to the Authors merging some of the paragraphs in the manuscript. It will improve the visual quality of the manuscript. 

Response 1: The paragraphs were merged in two places in the Introduction part of the manuscript: 

The concept of circular economy highlights the minimisation of disposed waste by recycling it in various industrial processes. The recycling of bottom ash from the incineration of municipal solid waste (MSW) in cement-based materials is one of the trends in the development of circular economy. The use of MSW has a unique potential to support sustainability while preserving natural resources. Greenhouse gas (GHG) emissions in cement manufacture are very high, accounting for about 8% of the global carbon dioxide (CO2) emitted [1, 2]. Cement industry contributes to more than 85% of CO2 and GHG emissions [3]. The use of bottom ash can reduce the cement content in concrete and at the same time moderate the environmental pollution and CO2 emissions.

Great attention has been paid to the use of bottom ash as a secondary raw material for the production of Portland cement clinker [20-22]. One of the main reasons for using MSW bottom ash in cement clinker manufacturing process is that chemical composition of bottom ash is similar to that of the raw cement clinker materials, including lime, silica, alumina, iron oxide/hematite, and calcium sulphate [23]. Municipal solid waste incineration ash can partially replace Portland cement in concrete. Zhang et al. [24] tested the replacement of cement with 15% bottom ash in concrete mixes. The results showed that concrete specimens had much better strength and durability values when activated by carbonation. Dry-cast concrete with 20% cement replaced by bottom ash showed 18% higher strength [25]. Cheng et al. [26] found that the optimal ash content in concrete was between 10% and 15%. Within this range, the ash not only improves the durability of concrete, but also ensures that concrete strength meets the design requirements. Other researchers [27] recommend replacing up to 15% of cement with ash to achieve low strength concrete. Although bottom ash has a porous structure, good pozzolanic properties of the ash were observed after grinding and thus bottom ash can be used as supplementary cementitious materials.

Point 2: Please, double check my previous comment 14. It hasn’t been addressed. There is a mistake in units of measurements. One cannot obtain open porosity (Pa, %) when water absorption (W, %) is multiplied by density (ρ, g/cm3). 

Response 2: Thank you for the comments. The density of the liquid in which the samples were soaked, that is to say the density of the water, must be assessed. It was corrected. The adjusted places in the article are marked in yellow.